# The Role of Process-Directing Agents on Enamel Lesion Remineralization: Fluoride Boosters

**DOI:** 10.3390/biomimetics7020054

**Published:** 2022-04-28

**Authors:** Hamid Nurrohman, Logan Carter, Noah Barnes, Syeda Zehra, Vineet Singh, Jinhui Tao, Sally J. Marshall, Grayson W. Marshall

**Affiliations:** 1Missouri School of Dentistry and Oral Health, A.T. Still University, Kirksville, MO 63501, USA; sa200719@atsu.edu (L.C.); sa200611@atsu.edu (N.B.); sa202998@atsu.edu (S.Z.); vsingh@atsu.edu (V.S.); 2Preventive and Restorative Dental Sciences, University of California, San Francisco, CA 94143, USA; sally.marshall@ucsf.edu (S.J.M.); gw.marshall@ucsf.edu (G.W.M.); 3Physical and Computational Sciences Directorate, Pacific Northwest National Laboratory, Richland, WA 99352, USA; jinhui.tao@pnnl.gov

**Keywords:** PILP, process-directing agent, polyaspartic acid, osteopontin, bacteria-induced enamel demineralization

## Abstract

The aim of this study was to investigate the effects of two process-directing agents (polyaspartic acid and osteopontin) used in a polymer-induced liquid-precursor (PILP) process on the remineralization of bacteria-induced enamel demineralization. Enamel demineralization lesions (depths of about 180–200 µm) were created and exposed to Streptococcus mutans, cultured with a 10% sucrose solution for 21 days, and remineralized using a PILP process (pH = 7.4, 14 days) with a calcium phosphate solution containing either polyaspartic acid or osteopontin in the presence or absence of fluoride (0.5 ppm). The specimens were examined under scanning electron microscopy. The fluoride was successfully incorporated into the PILP remineralization process for both polyaspartic acid and osteopontin. When the fluoride was added to the PILP remineralization solution, there was more uniform remineralization throughout the lesion than with either polyaspartic acid or osteopontin alone. However, in the absence of these process-directing agents, fluoride alone showed less remineralization with the formation of a predominantly surface-only layer. The PILP remineralization process relies on the ability of process-directing agents to stabilize calcium phosphate ions and holds promise for enamel lesion remineralization, and these agents, in the presence of fluoride, seem to play an important role as a booster or supplement in the continuation of remineralization by reducing the mineral gains at the surface layer.

## 1. Introduction

Dental caries continues to be a highly prevalent global disease [1,2]. Carious lesions are caused by the organic acids produced by bacteria in the biofilm on tooth structure, enamel, and dentin. A goal of modern dentistry is to manage caries lesions noninvasively through remineralization. Fluoride-mediated remineralization is the cornerstone of current caries management philosophies [1,2,3]. However, fluoride alone is not sufficient in highly cariogenic oral environments. Thus, it is important to boost the remineralizing efficacy of fluoride [3].

We have suggested a proof-of-concept remineralization method based on formation of a polymer-induced liquid-precursor (PILP) process [4,5,6,7,8,9]. In the PILP process, a process-directing agent (such as polyaspartic acid—pASP) serves as a mimic for the highly acidic noncollagenous proteins (NCPs) found in mineralized tissues that are believed to play pivotal roles in collagen type I mineralization [9]. Our prior studies have shown significant success in remineralizing a variety of organic matrices, including rat tail tendon and dentin matrices [4,5,6,7,8,9]. However, the effectiveness of the PILP process has not been investigated for enamel lesion remineralization. Enamel and dentin contain apatite mineral crystals in an organic matrix. The structures and compositions vary between the tissues, but both form from protein interactions and self-assembly with crystallization controlled by the organic matrix [10].

One NCP that has been heavily studied is osteopontin (OPN), a member of the SIBLING (small integrin-binding ligand, N-linked glycoprotein) family of proteins. We sought to determine whether or not OPN can be used as a process-directing agent for the remineralization of enamel lesions under the same conditions of pASP used in the PILP process on collagen. Given that OPN has a domain with 8 to 10 consecutive aspartic acid residues, along with many phosphorylated residues [11,12,13,14,15], it might be reasonable to expect similar ion interactions with pASP. Moreover, phosphorylation was proved to play important roles in the regulation of calcium phosphate mineralization [10].

Based on these considerations, the purpose of this study was to (1) investigate the effect of two process-directing agents (pASP and OPN) used in the PILP process on the remineralization of bacteria-induced enamel demineralization and to (2) determine if pASP and OPN can act also as boosters or supplements to improve the remineralizing efficacy of fluoride. The mineralization aspect of the specimen was examined by the micromorphological appearance of samples using scanning electron microscopy (SEM).

## 2. Materials and Methods

### 2.1. Tooth Preparation

The ATSU institutional review board deemed this study exempt under CFR Section: 45CFR46.101 (b) (4). The Institutional Review Board of A.T. Still University approved the protocol used to obtain informed consent for the collection of thirty noncarious human third molars. The teeth were used within 1 month after extraction and storage in water at 4 °C. A low-speed diamond saw (Isomet; Buehler, Lake Buff, IL, USA) was used to section the crowns mesiodistally after removal of the roots. An optical microscope (Olympus CX33, Tokyo, Japan) at 10x magnification was used to exclude any surface defects or decalcified areas. To prepare the mid-coronal buccal enamel surfaces for the demineralization–remineralization process, crowns with buccal surfaces were polished with flour pumice, cleaned with 70% ethanol prior to inoculation for biofilm growth, and rinsed with sterile deionized water. Each specimen surface was sealed with nail varnish (Revlon, New York, NY, USA) except for a 3 × 3-mm window to prevent demineralization–remineralization cycles. 

Thirty prepared enamel discs were randomly divided into six groups (*n* = 5): (1) bacteria-induced enamel demineralization (BIED-control); (2) bacteria-induced enamel demineralization followed by remineralization in a PILP solution containing fluoride and no process-directing agent (BIED-REM_F_); (3) bacteria-induced enamel demineralization followed by remineralization in a PILP solution containing pASP as a process-directing agent (BIED-REM_pASP_); (4) bacteria-induced enamel demineralization followed by remineralization in a PILP solution containing pASP as a process-directing agent and fluoride BIED-REM_pASP-F_); (5) bacteria-induced enamel demineralization followed by remineralization in a PILP solution containing OPN as a process-directing agent (BIED-REM_OPN_); and (6) bacteria-induced enamel demineralization followed by remineralization in a PILP solution containing OPN as a process-directing agent and fluoride (BIED-REM_OPN-F_). 

### 2.2. Bacteria-Induced Enamel Demineralization

The biofilm was grown using *Streptococcus mutans* (ATCC 25175) as follows. A plastic Petri dish filled with 5 mL Brain Heart Infusion medium (BHI, Difco, Sparks, MD, USA) supplemented with 10% freshly prepared sucrose solution was used for each enamel disc. A pilot study indicated that incubation of the Petri dish for 21 days at 37 °C as a static culture would create 180–200 µm deep demineralized lesions [4,5,6,7,8,9]. The medium was replaced with prewarmed fresh BHI with 10% sucrose every 24 h. Microscopic inspection of the removed media was used to check for obvious contamination. Following demineralization, the biofilm was removed from each enamel disk using a cotton tip and vigorous rinsing with deionized water.

### 2.3. Remineralization Experiments

An amount of 40 mL of remineralization solution at 37 °C with continuous shaking for 14 days was used for each specimen after lesion formation [5,8,9]. A normal PILP remineralization solution with 50 mM Tris-buffer with 0.9% NaCl, 0.02% NaN_3_, 4.5 mM CaCl_2_, and 2.1 mM K_2_HPO_4_ at a pH of 7.4 was used. The process-directing agent used was either bovine-milk-derived osteopontin (OPN; Lacprodan^®^ OPN-10, Arla Foods Ingredients Group P/S) or 27 kDa poly-L-aspartic acid (pASP; Alamanda Polymers). The pH of the PILP remineralization process was adjusted to 7.4 by the addition of a NaOH solution at room temperature to maintain a pH of 7.4 in the PILP remineralization solution. According to our previous study [7], fluoride (0.5 ppm) was used in some of the PILP groups.

After embedding in epoxy resin (Epoxicure, Buehler) overnight, each demineralized–remineralized disc was cut (low-speed water-cooled diamond saw, Isomet, Buehler) perpendicular to the surface, exposing the lesion structure ranging from the most demineralized outer portion through the lesion into sound dentin. SiC abrasive papers from 320 to 1200 grits and aqueous diamond suspensions of 3.0, 1.0, and 0.25 μm particle sizes were used to polish the specimens. The micromorphological aspects of the specimens were evaluated using a tabletop scanning electron microscope (SEM, TM3000, Hitachi-High Technology, Tokyo, Japan) at an acceleration voltage of 15 kV. 

## 3. Results

The typical morphologies after demineralization–remineralization are shown in Figure 1, Figure 2, Figure 3, Figure 4, Figure 5 and Figure 6. An outer lesion (OL) created by mineral loss due to biofilm colonization was observed in the BIED-control group (Figure 1). The depth of the OLs ranged from 180 to 200 µm in the control group. These images provided baseline information on the depth of the demineralization and the extent of apatite dissolution in human enamel.

For the BIED-REM_F_ specimen (Figure 2), an approximately 30 µm thick electron-dense zone was observed at the outer zone of the enamel lesion. At a higher magnification, it was noted that the zone consisted of densely arranged grain-like crystallites (Figure 2B). Sparsely arranged electron-dense regions were observed in the inner zone of the enamel lesion could be distinguished. This surface-only remineralization improved neither the aesthetics nor the structural properties of the subsurface lesion.

For the BIED-REM_pASP_ group (Figure 3), an electron-dense zone through the lesion depth was observed. Interestingly, the BIED-REM_pASP-F_ group (Figure 4) had higher electron density at the inner zone of the enamel lesion compared with that of the BIED-REM_pASP_ group. 

For the BIED-REM_OPN_ group (Figure 5), thick electron-dense zones were seen at both the outer and, mostly, the inner zones of the enamel lesion. 

For the BIED-REM_OPN-F_ group (Figure 6), the morphology was almost similar to the corresponding region of the BIED-REM_OPN_ group. Surprisingly, an examination under higher magnification revealed a few sparse, grain-like structures at the middle of the enamel lesions. 

## 4. Discussion

This study sought to evaluate the roles of process-directing agents (pASP and OPN) and fluoride in controlling remineralization and structural recovery of a bacteria-induced enamel demineralization model. Fluoride remains the gold standard for arresting caries lesions with multiple systematic reviews confirming the role of fluoride in preventing dental caries [16,17]. However, while the dose of fluoride is a key factor in dental caries treatment, high concentrations of fluoride could raise safety concerns and potentially increase the risk of developing dental fluorosis [18]. Moreover, a low concentration of fluoride is not adequate in highly cariogenic oral environments [1]. New remineralization techniques using lower fluoride levels would benefit many people of concern, including at-risk population groups such as xerostomia patients [3]. In addition, new non-fluoride remineralization processes potentially may allow the design of dental products with lower fluoride concentrations, addressing any safety concerns about high fluoride concentrations.

For reference, the amount of fluoride used in our previous study in remineralization solutions was 0–200 ppm [7]. Interestingly, as the fluoride concentration increased, the apatite crystals became larger and formed on the fibril surfaces rather than the interior, as seen when low levels of fluoride were added; this was particularly evident in the 0.5 ppm group [7]. Therefore, a low fluoride concentration was used (0.5 ppm) in this study, which also decreased risks associated with fluoride use (fluorosis) and toxicity. In this study, without process-directing agents, the combination of calcium and phosphate ions with fluoride ions could result in predominantly surface-only remineralization of demineralized enamel lesions (Figure 1). To improve the properties of a deeper lesion, subsurface mineral gain is needed, not just surface-only remineralization [2]. 

The PILP process has been shown to produce stabilized amorphous mineral precursors, and it is reasonable that they would enter the porosities of an enamel subsurface lesion and diffuse into the bulk of the subsurface lesion [19]. Further evidence for enamel remineralization was provided in a recent study using the PILP method on demineralized sections of mouse enamel [20]. This in vitro mouse model suggested that the prosses direction agent used in a PILP system mimicked the activity of enamelin to interact with the amelogenin nanoribbon scaffolds and also provided a method to grow apatite in demineralized enamel [20,21]. It is important to note that, when a process direction agent was provided with fluoride, SEM images showed improved remineralization (Figure 4 and Figure 6). Thus, the current study did not aim to develop a new strategy as a substitute for fluoride but identified some consistent evidence derived from the use of biomimetic strategies to support their potential use to boost the effect of fluoride. This finding is in line with our previous study that showed enhanced silver diamine fluoride therapy using the PILP process [5]. 

This study presented a guided enamel remineralization that holds great promise for remineralization with a low dose of fluoride. The casein phosphopeptide-amorphous calcium phosphate (CPP-ACP) non-fluoride remineralizing process was developed and studied extensively. It involved the tryptic digestion of milk caseinate to produce multiphosphorylated casein phosphopeptides (CPPs), increasing the milk protein’s solubility and ability to stabilize calcium phosphate complex ions [2]. Some randomized clinical trials and systematic reviews have suggested that CPP-ACP has led to significant remineralization and caries prevention [22,23], but conflicting results have been found by others, concluding that the evidence to support its long-term remineralizing [24,25] or synergistic effect with fluoride is absent [26].

## 5. Conclusions

A goal of modern dentistry is the noninvasive management of caries lesions involving remineralization to repair enamel lesions with fluorapatite or fluoride-substituted hydroxyapatite. In this study, we were able to successfully incorporate fluoride ions into the remineralization of biologically relevant bacteria-induced enamel demineralization with the PILP process. The process-directing agents utilized in the PILP process to stabilize and deliver mineral ions and the control remineralization had the potential to boost the performance of fluoride. Clinically, the development of fluoroapatite is important since it has greater stability than hydroxyapatite in an acidic environment, minimizing the chances of cavity formation. The mineral levels, nanomechanical properties of the repaired tissue, and the crystallization mechanism will be determined in future research. 

## Figures and Tables

**Figure 1 biomimetics-07-00054-f001:**
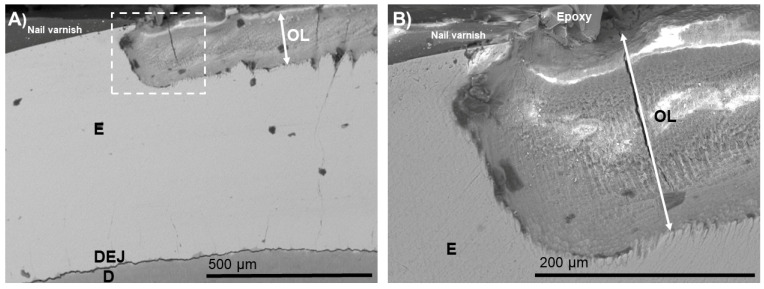
(**A**) SEM images of cross-sections of the bacteria-induced enamel demineralization (BIED-control) with nail varnish shown protecting the unexposed surface. (**B**) High-magnification SEM image of (**A**) showing the area under the dotted outline rectangle. E: enamel; D: dentin; DEJ: dentin–enamel junction; OL: outer lesion.

**Figure 2 biomimetics-07-00054-f002:**
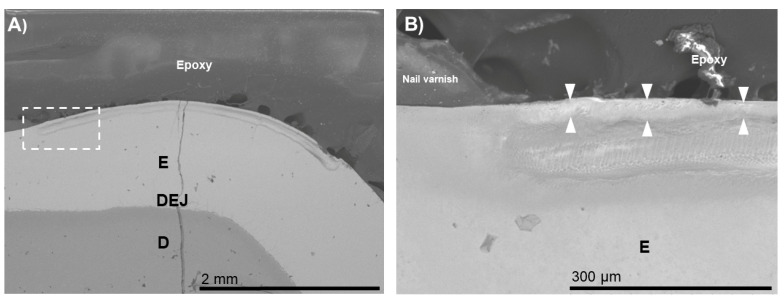
SEM images of the specimens in the bacteria-induced enamel demineralization followed by remineralization in PILP solution containing fluoride and no process-directing agent (BIED-REM_F_ group). (**B**) High-magnification SEM image of (**A**) showing the area under the dotted outline rectangle revealed thick electron-dense zone at the outer zone of the enamel lesion (area between the triangles). E: enamel; D: dentin; DEJ: dentin–enamel junction.

**Figure 3 biomimetics-07-00054-f003:**
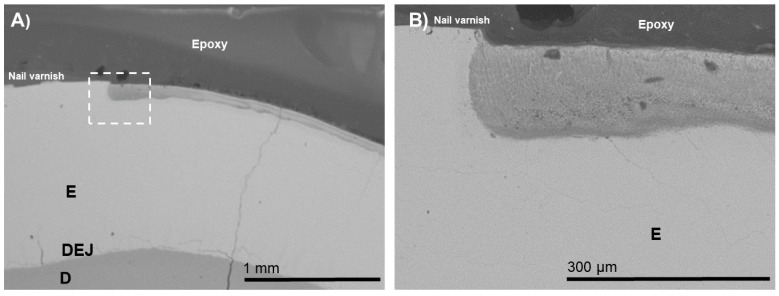
SEM images of the specimens in the bacteria-induced enamel demineralization followed by remineralization in PILP solution containing pASP as process-directing agent (BIED-REM_pASP_ group). (**B**) High-magnification SEM image of (**A**) showing the area under the dotted outline rectangle. E: enamel; D: dentin; DEJ: dentin–enamel junction.

**Figure 4 biomimetics-07-00054-f004:**
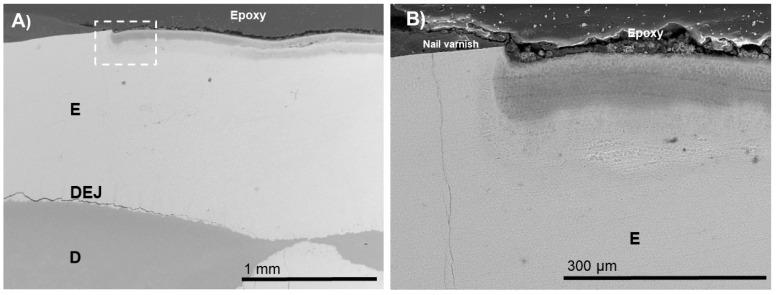
SEM images of the specimens in the bacteria-induced enamel demineralization followed by remineralization in PILP solution containing pASP as process-directing agent and fluoride (BIED-REM_pASP_-_F_ group). (**B**) High-magnification SEM image of (**A**) showing the area under the dotted outline rectangle. E: enamel; D: dentin; DEJ: dentin–enamel junction.

**Figure 5 biomimetics-07-00054-f005:**
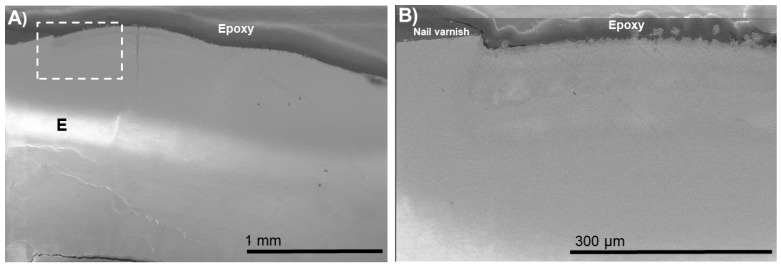
SEM images of the specimens in the bacteria-induced enamel demineralization followed by remineralization in PILP solution containing OPN as process-directing agent (BIED-REM_OPN_ group). (**B**) High-magnification SEM image of (**A**) showing the area under the dotted outline rectangle. E: enamel.

**Figure 6 biomimetics-07-00054-f006:**
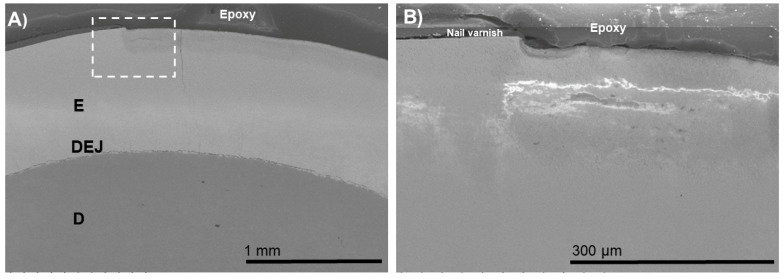
SEM images of the specimens in the bacteria-induced enamel demineralization followed by remineralization in PILP solution containing OPN as process-directing agent and fluoride (BIED-REM_OPN-F_ group). (**B**) High-magnification SEM image of (**A**) showing the area under the dotted outline rectangle. E: enamel; D: dentin; DEJ: dentin–enamel junction.

## Data Availability

Not applicable.

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
