# Peer review of "The Role of Process-Directing Agents on Enamel Lesion Remineralization: Fluoride Boosters"

_biomimetics, 2022, doi:10.3390/biomimetics7020054_

Round 1

Reviewer 1 Report

Thank you for giving this interesting chance to review. And I really enjoyed your research for the remineralization process on enamel. However, I should have raised some issue on this experiment.

First, why did you place the specimens in remineralization solution for 14 days? Is there any reference or just for until you can find the good result?

Second, I believed that you showed the wonderful figures to show how it worked. On the other hands, I think that the readers would like to know how much they worked in the numbers (quantitatively).

I hope that this could help your works improved. I appreciate your work.

Reviewer 2 Report

In this manuscript the authors employ the PILP approach to examine the effect of fluoride and osteopontin on enamel remineralization. While the topic of the study is important, the experimental design and results are lacking in certain aspects. My key comments are noted below:

  1. The authors employ the PILP method which is mostly used for collagen mineralization. Enamel is a non-collagenous tissue and solutions like simulated body fluid (SBF) or simulated dental fluid (SDF) are better suited to study enamel mineralization. (E.g. Farzadi A et al, Mater. Sci.& Eng, C, 2019; Shin K et al, Tissue Eng A, 2017)
  2. The SEM images presented are low magnification. The morphology of mineral deposits cannot be ascertained from these images. Higher magnification along with energy dispersive spectroscopy should be presented to confirm the presence of in-vitro mineralization.
  3. The data presented is only qualitative. Quantification of mineral deposits (e.g. composition, thickness morphology, density etc should be presented). 
  4. The amount of fluoride added to remineralization solutions is not mentioned in the Methods section. It is briefly mentioned in the Discussion. The rationale for selecting this concentration is not clear.
  5. What do the authors mean by de-remineralization process? This term needs to be defined.

Reviewer 3 Report

This manuscript intends to evaluate two process-directing agents (pASP or OPN) used in the PILP process on the remineralization of bacteria-induced enamel demineralization; and 2) determine if pASP and OPN can act also as booster/supplements to improve the remineralizing efficacy of fluoride. 

I think the text was very well written and the idea is very good. However, the method used did not seem sufficient to prove that there was remineralization of the lesion. Furthermore, the formed lesion is more suitable for an erosion process than for a carious enamel lesion. There was erosion of the surface area of ​​the lesion, exposing the subsurface lesion, which significantly compromises the effect of the parameters tested. It would be convenient for the text to focus on erosive lesions rather than caries.

There is great difficulty in producing enamel lesions without erosion, especially using S.m. biofilm. Time is a great ally and reducing exposure time to biofilm and adding Fluoride in the medium helps to reduce erosion.
I suggest that authors rethink the focus of the work, and if possible combine other methods (uCT, XRD, microhardness) to improve the acceptability and credibility of the article.

Round 2

Reviewer 1 Report

I appreciated the author's work and thank you for giving this chance to review. I believe that the all raised issues were resolved. I recommend the acceptance in present form.

Reviewer 2 Report

The authors have edited the manuscript in response to my comments.